

# Genome-wide identification and expression analysis of the trehalose-6-phosphate synthase (*TPS*) gene family in cucumber (*Cucumis sativus* L.)

Yuanyuan Dan, Yuan Niu, Chunlei Wang, Mei Yan and Weibiao Liao

College of Horticulture, Gansu Agricultural University, Lanzhou, China

## ABSTRACT

Trehalose-6-phosphate synthase (TPS) is significant in the growth, development and stress resistance of plants. We identified the cucumber *TPS* family and its physicochemical properties, domains, gene structures, evolutionary relationships, gene locations, *cis*-acting elements, conserved motifs, and expression patterns using bioinformatics. Our results uncovered seven *CsTPS* genes in the cucumber genome and named *CsTPS1–CsTPS7* according to their locations in the chromosomes. Seven *CsTPS* genes were randomly distributed in six cucumber chromosomes. Domain analysis showed that the TPS and TPP domains exist in all CsTPSs, and an additional hydrolase-3 domain exist in CsTPS3, CsTPS5 and CsTPS6. Phylogenetic analysis showed that TPS proteins from *Arabidopsis*, rice, soybean, and cucumber were divided into two subfamilies (Class I and Class II) and they were further divided into seven subgroups. TPS proteins from *Arabidopsis* and cucumber were grouped together, suggesting a close evolutionary relationship. Gene structure analysis indicated that most Class I genes contained 16–17 introns, while Class II genes (except *CsTPS7*) had two introns. Motif analysis showed that Class II genes had 10 complete conserved motifs, while Class I genes lacked motif 8 and motif 9. Furthermore, *CsTPS* genes possessed numerous *cis*-acting elements related to stress, hormone, and light response in the promoter regions. GO analysis indicated multiple functions for the CsTPS proteins. Expression analysis of *CsTPS* genes in different tissues found that they were expressed in roots, stems and leaves, with the highest expression levels in roots. The expression analysis of *CsTPSs* under different treatments showed that *CsTPS* genes may participate in the response to abiotic stress, plant hormones and sugar treatments.

Corresponding author
Weibiao Liao, liaowb@gsau.edu.cn

## INTRODUCTION

The growth and development of plants in agriculture are often affected by various adverse conditions, including submergence, drought, low or high temperatures, and saline and alkaline soils (*Zhu, 2016*). These adverse conditions may dehydrate plants, which reduces the photosynthesis rate (*Baninasab, 2010*), promotes the production of reactive oxygen

species (ROS), and damages the cell membrane (*Deinlein et al., 2014*). Plants have produced a series of physiological mechanisms over time to protect themselves from adversity. For example, under stress conditions, plants express stress-related genes to produce stress proteins, including the heat shock protein (HSP), low temperature-induced protein, osmoregulatory protein, and the pathogenesis-related protein (PR) (*Hightower, 1991*). Plants may also accumulate osmotic adjustment substances (*Ashraf & Foolad, 2007*) under unfavorable conditions, including inorganic ions ($Na^+$, $K^+$ and $Cl^-$), proline, betaine, abscisic acid (*Kim, 2012*) and sugars (sucrose, fructose and trehalose) to protect the integrity of the membrane structure.

Trehalose (α-D-glucopyranosyl-1, 1-α-D-glucopyranoside) (*Lunn et al., 2014*) is a non-reducing disaccharide composed of two molecules of glucose (*Elbein et al., 2003*). In higher plants, trehalose is synthesized through the catalysis of two enzymes: trehalose-6-phosphate synthase (TPS) and trehalose-6-phosphate phosphatase (TPP). TPS first catalyzes UDP-glucose and glucose-6-phosphate to produce trehalose-6-phosphate (T6P) and UDP. Then, TPP dephosphorylates trehalose-6-phosphate to produce trehalose (*Goddijn & Van Dun, 1999*). Trehalose is widely found in plants and plays a specific role in plant growth, development, and resistance to stress. Compost treated with trehalose has been used to cultivate quinoa plants to increase their growth and yield (*Abdallah et al., 2020*), and soaking rice seeds with trehalose may relieve salt stress (*Abdallah, Abdelgawad & El-bassiouny, 2016*). Trehalose may form Cd-Trehalose chelate with cadmium (Cd) to alleviate the damage of cadmium stress in rice (*Wang et al., 2020*). The expression of the *TPS* gene and the accumulation of trehalose has been shown to increase when wheat are exposed to drought (*El-Bashiti et al., 2005*).

TPS plays a vital role in trehalose metabolism and stress resistance in plants (*Yang et al., 2012*). Previous studies have shown that light quality may affect the growth and phase transition by influencing the TPS1-T6P signaling pathway in tomatoes (*Chen & Lou, 2017*). The overexpression of the *TPS1* gene in rice and potato enhanced their stress resistance (*Li et al., 2011*; *Kondrák et al., 2011*). The overexpression of the *TPS11* gene in wheat improved cold resistance in *Arabidopsis* (*Liu et al., 2019b*); *TPS1* played an important role in the embryogenesis, post-embryonic growth, and development in *Arabidopsis* (*Fichtner et al., 2020*). Studies have shown that TPS affected development and metabolic processes by altering T6P levels (*Lunn et al., 2014*)

T6P is an intermediary in trehalose biosynthesis with a vital role in plant growth and development (*Yadav et al., 2014*; *Zhang et al., 2015*). T6P serves as a sugar-signaling molecule in *Arabidopsis* that coordinates the hypocotyl elongation mediated by high temperature and the availability of endogenous sugar (*Geonhee et al., 2019*). The accumulation of T6P inhibited the growth of *Arabidopsis* seedlings mediated by trehalose (*Schluepmann et al., 2003*). The T6P signaling pathway played an important role in the flowering of the *Arabidopsis* leaf and stem meristems (*Wahl et al., 2013*).

T6P was also involved in regulating the use and distribution of sucrose, coordinating source-sink relationships, the efficient use of carbohydrates (*Schluepmann et al., 2003*), and improving crop yield (*Paul, Watson & Griffiths, 2020*). Research on cucumber fruit has shown that there was a strong correlation between T6P and sucrose (*Zhang et al.,*
*2015*). The Tre6P:sucrose ratio could maintain sucrose levels within a range that is appropriate for the cell type and developmental stage of plants (*Yadav et al., 2014*). Varying T6P levels and sugar signaling through chemical intervention significantly impacted crop yield and resilience (*Griffiths et al., 2016*; *Smeekens, 2017*).

The overexpression of rice *TPP* genes in maize ears under well-watered or drought conditions reduced the level of T6P, increased the level of sucrose, and improved yield (*Nuccio et al., 2015*). A study of *Arabidopsis* indicated that many growth and developmental defects were due to T6P rather than trehalose (*Schluepmann et al., 2003*).

Cucumber (*Cucumis sativus* L.) is a widely cultivated crop with high nutritional value. Its growth and development are easily affected by adverse condition, especially salt stress (*Miao et al., 2020*). We sought to identify members of the *TPS* family in cucumber using bioinformatics methods. We analyzed the gene structure and location, motif distribution and composition, evolutionary relationship and expression patterns. We hope that our work supports future functional research of the *TPS* family in cucumber plants.

## MATERIALS AND METHODS

### Genome-wide identification and bioinformatics analysis

We downloaded the whole genome in the gff, cds, pep and FASTA file format from the EnsemblPlants- Cucumber genome database (ASM407v2) (http://plants.ensembl.org/index.html) (*Li et al., 2019b*). The TPS (Glyco-transf-20, PF00982) and TPP (Trehalose_PPase, PF02358) domains' hidden Markov Models (HMM) were downloaded from the Pfam database (http://pfam.xfam.org) (*Liu et al., 2019a*). HMMsearch software and the TPS and TPP's HMMs were used to search all possible TPS candidate sequences containing typical TPS and TPP domains under a Linux system (*Chen et al., 2019*). Pfam (*Liu et al., 2019a*) and NCBI-CDD (https://www.ncbi.nlm.nih.gov/Structure/bwrpsb/bwrpsb.cgi) (*Yan, Li & Zhao, 2019*) databases were used to manually confirm that the candidate sequences had complete TPS and TPP domains. The remaining genes were subsequently identified as members of the cucumber *TPS* family and were named according to their location on the cucumber's chromosome.

The length of open reading frames (ORFs) of cucumber *TPS* genes was predicted using the NCBI-ORFfinder (https://www.ncbi.nlm.nih.gov/orffinder/) website (*Xie et al., 2015*). The physicochemical properties and subcellular locations of the cucumber TPS proteins were forecast using Protparam (https://web.expasy.org/protparam/) (*Zhang et al., 2019*) and the Cell-PLoc2.0 (http://www.csbio.sjtu.edu.cn/bioinf/plant-multi/) website, respectively.

The cucumber TPS proteins' secondary structures were determined using the PRABI (https://npsa-prabi.ibcp.fr/cgi-bin/npsa_automat.pl?page=npsa_sopma.html) website (*Xu et al., 2016*).

### Phylogenetic analysis

The phylogenetic tree, containing seven cucumber, 11 rice (*Zang et al., 2011*), 11 *Arabidopsis* (*Yang et al., 2012*) and 20 soybean (*Xie, Wang & Huang, 2014*) TPS protein sequences, was constructed based on multiple sequence alignments using Fasttree software

and the maximum likelihood method. The bootstrap replication value was set as 1,000 and the other parameters remained constant. We used the evolview (https://evolgenius.info//evolview-v2/#login) website to improve the appearance of the evolutionary tree.

The TPS protein sequences of *Arabidopsis*, rice, and soybean were downloaded from TAIR (https://www.arabidopsis.org/) (*Song et al., 2019*), the Rice Genome Annotation Project (http://rice.plantbiology.msu.edu/) (*Zang et al., 2011*), and the Phytozome (https://phytozome.jgi.doe.gov) (*Yue et al., 2019*) database, respectively (Supplemental File 1).

## Gene structure, chromosomal location and *cis*-acting element analysis

The GSDS2.0 (http://gsds.gao-lab.org/index.php) (*Li et al., 2019a*) website was used to analyze the gene structure of cucumber *TPS* genes and to plot the exon-intron diagram. We used the TBtools software to combine the evolutionary tree with the gene structure diagram and Mapchart software was applied to visualize the location of the genes on the chromosomes. The 2kb sequences in the cucumber *TPS* genes' upstream region were screened as promoter sequences using Tbtools software. We used plantCARE (*Song et al., 2019*) (http://bioinformatics.psb.ugent.be/webtools/plantcare/html/) to investigate the *cis*-acting elements in promoter regions to study the roles of genes in stress and hormone responses.

## Conserved motifs analysis

We searched the MEME (http://meme-suite.org/) (*Liu et al., 2019c*) website for conserved motifs of cucumber TPS proteins. The maximum retrieval value for the motif was set to 10 and the other parameters were set to default. InterProScan software was used to annotate the retrieved motifs.

## GO annotation

We used the EggNOG mapper software (http://eggnog-mapper.embl.de/) to perform the gene ontology analysis. Cucumber TPS protein sequences were uploaded and *Arabidopsis* TPSs were used as the reference. GO analysis was categorized as: molecular function (MF), biological process (BP), and cellular component (CC).

## Plant materials, cultivation conditions and treatments

Cucumber seeds (*C. sativus* L. "Xin Chun 4") were germinated and grown in culture dishes with wet filter paper. Seedlings were transferred into hydroponic boxes once the cotyledon fully unfolded. The boxes contained Yamazaki cucumber nutrient solution and were placed in plant incubators at 25 °C, a light intensity of 200 $\mu mol.m^{-2}s^{-1}$, and a photoperiod of 14 h light/10 h dark (*Niu et al., 2019*). The nutrient solution was replaced every two days in order to maintain an adequate level of nutrients. Stress treatments were carried out when seedlings were at the two-leaf stage. Seedlings were grown in a 1/2 nutrient solution containing 8% (w/v) PEG, 50 mM NaCl, 1 $\mu$M IAA, 8% (w/v) $H_2O_2$, 50 mM sucrose
**Table 1 Primer sequence for qRT-PCR.**

| Gene name | Primer sequence (5′ to 3′) | |
|---|---|---|
| CsTPS1 | F: AAGTGGTGCTGTCAGGGTAAATCC | R: GCCCAGTAAGCAACATCGTGAGAG |
| CsTPS2 | F: AGCGTTGGTGGTTTAGTCAGTGC | R: TGCTTTCTCTAGGGCTCTCTGTCC |
| CsTPS3 | F: TGGGCTCGGAGAAGATGTGGAAG | R: GTCGGGACGCACTTGAATCGG |
| CsTPS4 | F: ACCCTTCCATCCCGATCAGAGC | R: TCCTTGGTCCTCAACTCCTTCTGG |
| CsTPS5 | F: AAGCCAAGGAATTGCTGGACCATC | R: TGCGACCAACCCTTTGCTTACTC |
| CsTPS6 | F: CTGTCATGCCGCAAACTTCAATCG | R: AAACTTTCACGCCCTCTTCCACTG |
| CsTPS7 | F: AGACGGTGTTGCTTGGTGTTGATG | R: ACAGCCTTCCCTTGCCACTTTG |
| CsActin | F: TGGACTCTGGTGATGGTGTTA | R: CAATGAGGGATGGCTGGAAAA |

and 50 mM mannitol, respectively, for drought, salt, IAA, $H_2O_2$, sucrose and mannitol treatments. The concentrations of these reagents were determined by a preliminary experiment. Whole seedlings were frozen with liquid nitrogen after treatment at 0, 6, 12 and 24 h, and were stored at −80 °C (Zhu et al., 2019). The roots, stems, and leaves of untreated seedlings were collected at the two-leaf stage and were stored at −80 °C to investigate the expression of cucumber *TPS* genes in the different tissues. Each treatment was performed with three biological replicates and each sample was collected from five cucumber seedlings.

## RNA extraction, reverse transcription and quantitative real-time PCR

The total RNA from different tissues and whole seedlings under different treatments were extracted using the MiniBEST Plant RNA Extraction Kit (TaKaRa, Dalian, China). The RNA concentration and purity were determined using the NaNo drop 1,000 spectrophotometer and agarose gel electrophoresis (Xie et al., 2018). The FastQuant first strand cDNA synthesis kit (TIANGEN, Beijing, China) was used for the synthesis of cDNA following the manufacturer's protocol. The SuperReal PreMix Plus kit (TIANGEN, Beijing, China) and a Roche LightCycler instrument were used for qRT-PCR. The reaction system of qRT-PCR was as follows: 10 μL 2×SuperReal PreMix Plus, 0.6 μL 10 μM forward primers, 0.6 μL 10 μM reverse primers, 2 μL cDNA and 6.8 μL RNase-free ddH$_2$O. The qRT-PCR procedure was as follow: 95 °C for 15 min and 40 cycles of 95 °C for 10 s and 60 °C for 20 s. *CsActin* was used as an internal reference gene (Zhou et al., 2017). The primers of the cucumber *TPS* genes and *CsActin* for qRT-PCR were designed and synthesized using Sangon Biotech online software (Table 1). Three technical replicates were performed for each reaction.

### Statistical analysis

The relative expression of the genes was calculated using the $2^{-\Delta\Delta Ct}$ method (Han et al., 2020). The expression of the *CsTPS* genes in the roots was used as a calibration sample to calculate their relative expressions in the stem and leaf (Zhang et al., 2014). The expression of *CsTPS* genes in untreated seedlings (0 h) was used as a calibration sample to calculate their relative expression at 6, 12 and 24 h. We adopted Duncan's ($p < 0.05$) method for significance analysis.

**Table 2 Fundamental information of *CsTPS* genes.**

| Gene | Gene ID | Gene locus | ORF(bp) | Amino acid | Molecular weight/KDa | pI | TPS domain location | TPP domain location | Subcellular Localization |
|------|---------|------------|---------|------------|----------------------|-----|---------------------|---------------------|--------------------------|
| *CsTPS1* | Csa_1G005560 | Chr1 | 2568 | 855 | 97.14 | 6.34 | 53-540 | 589-824 | Chloroplast. Vacuole. |
| *CsTPS2* | Csa_1G467060 | Chr1 | 2604 | 867 | 97.87 | 7.02 | 21-492 | 537-765 | Chloroplast. Vacuole. |
| *CsTPS3* | Csa_3G009420 | Chr3 | 2595 | 864 | 97.46 | 5.73 | 62-548 | 597-830 | Cytoplasm. Vacuole. |
| *CsTPS4* | Csa_4G622880 | Chr4 | 2787 | 928 | 105.15 | 6.24 | 94-561 | 606-834 | Cytoplasm. Vacuole. Chloroplast. |
| *CsTPS5* | Csa_5G602180 | Chr5 | 2583 | 860 | 97.23 | 6.25 | 58-545 | 594-829 | Cytoplasm. Vacuole. |
| *CsTPS6* | Csa_6G520240 | Chr6 | 2553 | 850 | 96.31 | 5.50 | 59-544 | 593-828 | Cytoplasm. Vacuole. |
| *CsTPS7* | Csa_7G049190 | Chr7 | 2916 | 971 | 110.33 | 6.85 | 174-659 | 708-943 | Chloroplast. Vacuole. |

# RESULTS

## Genome-wide identification of *TPS* family in cucumber

Seven *TPS* sequences were identified in the cucumber genome using bioinformatics methods. The Pfam and NCBI-CDD databases were used for domain analysis to further prove the reliability of these candidate sequences. Our results indicated that all seven sequences had a typical TPS domain and belonged to the cucumber *TPS* family. The cucumber *TPS* genes identified were named *CsTPS1-CsTPS7* according to their location in the cucumber chromosome (Table 2).

Domain analysis revealed that all cucumber TPS proteins contained a typical TPS (Glyco_transf_20; Pfam: PF00982) domain in the N-terminal and a TPP (Trehalose_PPase; Pfam: PF02358) domain in the C-terminal (Table 2). CsTPS3, CsTPS5, and CsTPS6 proteins all contained an additional Hydrolase-3 (Pfam: PF08282) domain (Supplemental File 2).

Physical and chemical property analysis revealed that the length of the open reading frame of the *CsTPS* genes was between 2,553 bp (*CsTPS6*) and 2,916 bp (*CsTPS7*); the length of cucumber TPS proteins was between 850 (CsTPS6) and 971 (CsTPS7) amino acids; and their molecular weight was between 96.31 KDa (CsTPS6) and 110.33 KDa (CsTPS7). The isoelectric point (pI) ranged from 5.50 (CsTPS6) to 7.02 (CsTPS2). CsTPS2 was the only alkalescence (pI > 7) in the cucumber TPS proteins and the rest were acidic (pI < 7) (Table 2).

The subcellular localization prediction indicated that cucumber *TPS* genes were mainly distributed in the vacuole, chloroplast, and cytoplasm (Table 2).

The analysis of the secondary structure showed that cucumber TPS proteins were composed of an α-helix, random coil, extended strand and a β-turn (Table 3).

## Phylogeny and gene structure analysis of *CsTPSs*

Seven cucumber TPS proteins were divided into two subfamilies, Class I and Class II, based on the results of previous studies (*Lunn, 2007*; *Mu et al., 2016*). Among the seven CsTPS proteins, CsTPS2 and CsTPS4 were classified as Class I and the remaining five proteins were classified as Class II (Fig. 1A). In addition, the exon-intron diagram revealed

**Table 3 The secondary structures of CsTPS proteins.**

| Protein | Alpha helix (%) | Beta turn (%) | Random coil (%) | Extended strand (%) |
|---|---|---|---|---|
| CsTPS1 | 42.81 | 4.21 | 35.79 | 17.19 |
| CsTPS2 | 42.68 | 5.19 | 35.87 | 16.26 |
| CsTPS3 | 43.63 | 5.44 | 34.26 | 16.67 |
| CsTPS4 | 43.97 | 5.39 | 37.07 | 13.58 |
| CsTPS5 | 41.98 | 5.35 | 35.23 | 17.44 |
| CsTPS6 | 43.06 | 5.18 | 34.82 | 16.94 |
| CsTPS7 | 39.03 | 5.46 | 36.05 | 19.46 |

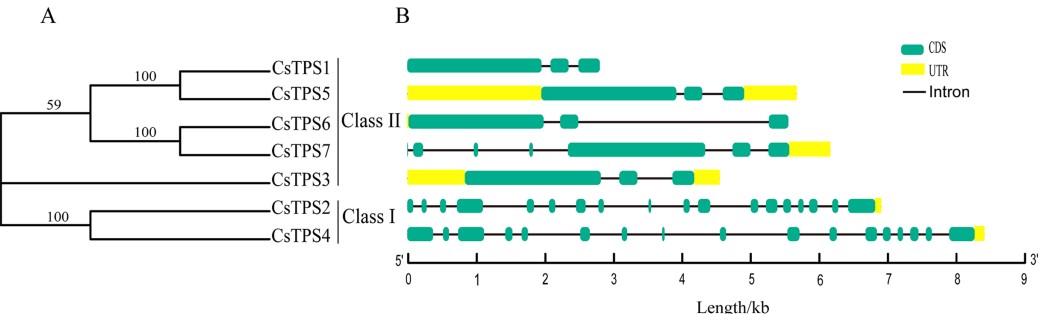

**Figure 1 Phylogenetic relationships and gene structures of CsTPSs.** (A) The evolutionary tree was built based on the full-length cucumber TPS protein sequences using Fasttree software. (B) The exon-intron diagram of cucumber *TPS* genes was mapped using GSDS2.0.

that two members (*CsTPS2* and *CsTPS4*) of Class I possessed 17 and 16 introns, respectively. In Class II, only *CsTPS7* had six introns, while the other members all contained two introns (Fig. 1B). We inferred that two subfamilies went through functional differentiation in the course of evolution (*Du et al., 2017*).

We aligned the full-length protein sequences of seven cucumber TPS proteins, 11 *Arabidopsis* TPS proteins (*Yang et al., 2012*), 11 rice TPS proteins (*Zang et al., 2011*) and 20 soybean TPS proteins (*Xie, Wang & Huang, 2014*) to establish a maximum likelihood phylogenetic tree (Fig. 2) to further investigate the evolutionary relationships of TPS family in various species. As previously described, the TPSs in cucumber, *Arabidopsis*, rice, and soybean may be differentiated into two subfamilies: Class I and Class II. The two subfamilies were further divided into seven subgroups: I-1, I-2, II-1, II-2, II-3, II-4 and II-5, based on the phylogenetic relationship with high bootstrap support (*Xie et al., 2015*). Subgroup I-1 contained nine members originating from *Arabidopsis* (1), rice (1), cucumber (2) and soybean (5), respectively. I-2 contained three members which all originated from *Arabidopsis*. II-1 contained nine members which were derived from *Arabidopsis* (2), rice (2), cucumber (1), and soybean (4), respectively. II-2 contained four members which derived from cucumber (1) and soybean (3), respectively. II-3 contained ten members which originated from *Arabidopsis* (3), rice (2), cucumber (1), and soybean (4), respectively. II-4 contained six members which originated from *Arabidopsis* (1), rice (3), and soybean (2), respectively. II-5 contained eight members which derived

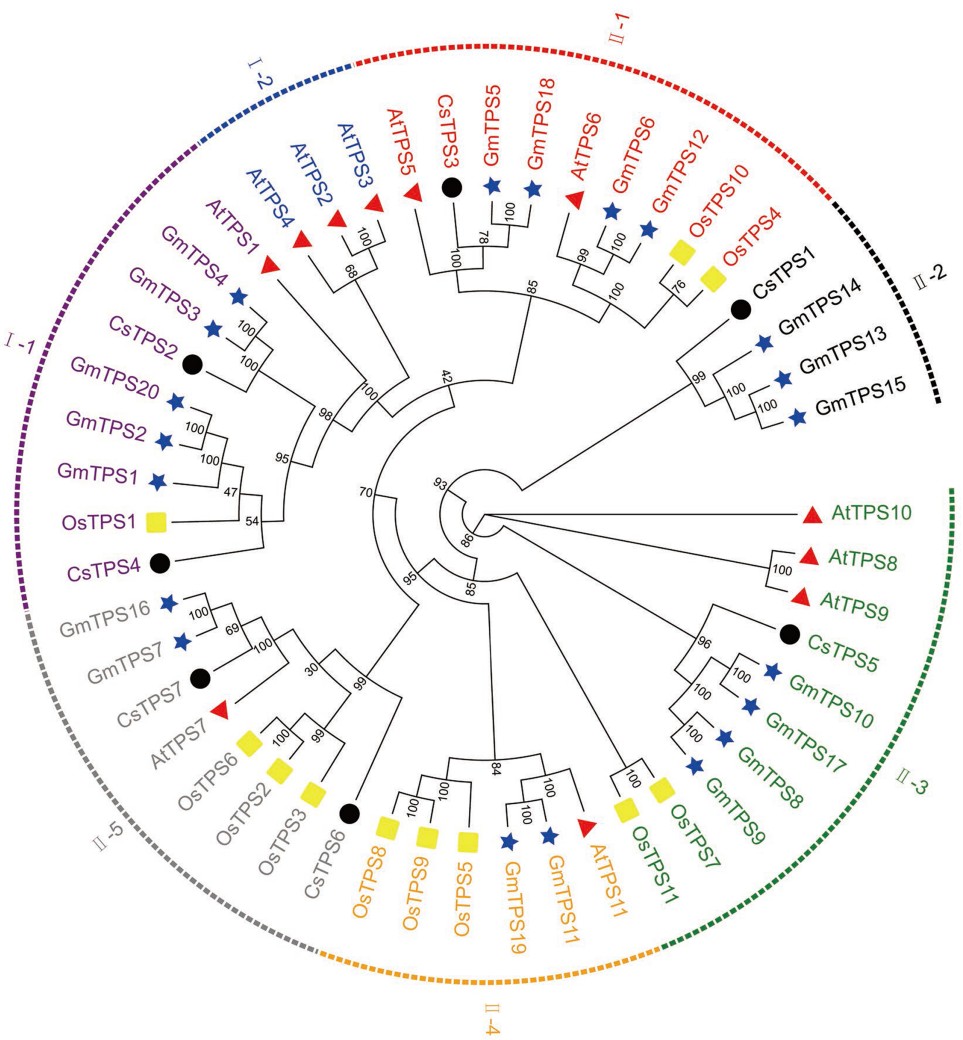

**Figure 2 Evolutionary relationships of *TPS* family in various species.** A phylogenetic tree containing seven cucumber, 11 rice (Os), 11 *Arabidopsis* (At), and 20 soybean (Gm) TPS proteins was constructed using the maximum likelihood method (*Cheng et al., 2018*). The seven subgroups are colored differently. The four differently-colored shapes represent TPS proteins from four species. The black circle, yellow rectangle, red triangle, and blue star represent cucumber, rice, *Arabidopsis*, and soybean TPS proteins, respectively.

from *Arabidopsis* (1), rice (3), cucumber (2), and soybean (2), respectively. With the exception of I-2, II-2 and II-4, the other subgroups contained at least one member of four species. Our results showed that some of the TPS proteins of cucumber and soybean were divided into the same subgroup, indicating that they were closely related. In addition, previous studies have shown that most of *TPS* genes in the Class I had 16 introns, while *TPS* genes in the Class II possessed 2 introns (*Yang et al., 2012*). When combined with the analysis of the cucumber *TPS* gene structure, we can speculate that the genes of the same group were close to each other during the evolution process, while the genes of different groups were far away from each other during the evolution process (*Xie et al., 2015*).

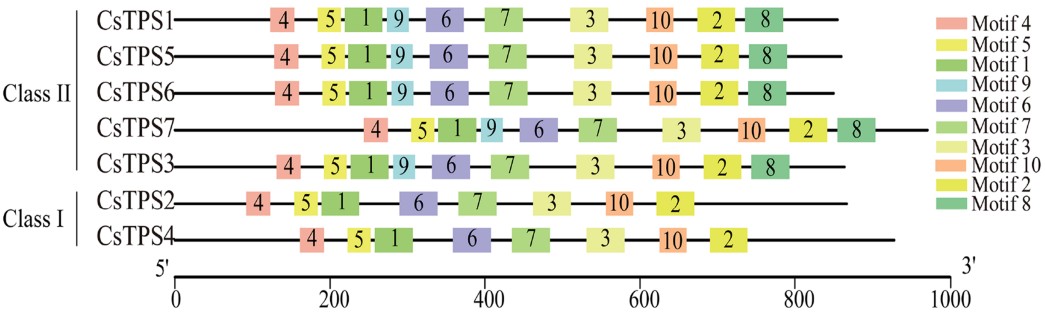

**Figure 3 The motif composition and distribution of cucumber TPS proteins.** Colored boxes represent different conserved motifs.          

**Table 4 Details of the 10 conserved motifs of cucumber TPS proteins.**

| Motif | Width (aa) | Motif Sequence | Annotation |
|---|---|---|---|
| Motif 1 | 50 | GFFLHSPFPSSEIYRTLPVRDEJLRALLNADLIGFHTFDYARHFLSCCSR | Glyco-transf-20 |
| Motif 2 | 50 | WIQIAEPVMKLYTEATDGSHIETKESALVWHYQDADPDFGSCQAKELLDH | Trehalose_PPase |
| Motif 3 | 50 | KQLRHEKHYRYVSTHDVAYWSRSFLQDLERACRDHYRRCWGIGFGLGFR | — |
| Motif 4 | 32 | FKCIPTFLPPEJLKQFYHGFCKQHLWPLFHYM | Glyco-transf-20 |
| Motif 5 | 31 | VVEVINPEDDYVWIHDYHLMVLPTFLRKRFN | Glyco-transf-20 |
| Motif 6 | 50 | FKGKKVJLGVDDLDIFKGINLKLLAFEQLLRQHPKWRGKAVLVQIANPAR | Glyco-transf-20 |
| Motif 7 | 50 | PGYEPIVLJDRPVPFHERIAYYAIAECCJVTAVRDGMNLVPYEYVVCRQG | Glyco-transf-20 |
| Motif 8 | 50 | KSGQHIVEVKPQGVSKGLVAEKILSSMAESGKLPDFVLCIGDDRSDEDMF | Trehalose_PPase |
| Motif 9 | 29 | YQSKRGYIGLEYYGRTVGIKILPVGIHMG | — |
| Motif 10 | 36 | EVISILNTLCDDPKNTVFIVSGRGRSSLGDWFGPCE | Trehalose_PPase |

## Conserved motifs of cucumber TPS proteins

We used the MEME online website to study the characteristic regions of cucumber TPS proteins. We searched 10 conserved motifs in cucumber TPS proteins (Fig. 3). The lengths of these conserved motifs were between 29 and 50 amino acids and the sequence information of these 10 conserved motifs is listed in Table 4. Cucumber TPS proteins classified into the same subfamily in the evolutionary tree possessed a similar or identical motif composition (*Ou et al., 2018*). For instance, members of Class II possessed all 10 conserved motifs, while members (CsTPS2 and CsTPS4) of Class I lacked motifs 8 and 9. Motifs 4, 5, 1, 9, 6, 7 and 3 together constituted the TPS domain and motifs 10, 2 and 8 constituted the TPP domain, according to annotation. These results further supported the reliability of the phylogenetic classification of cucumber TPS proteins and indicated that the two subfamilies may have produced a functional difference during evolution (*Xie et al., 2018*).

## Chromosomal location of cucumber *TPS* genes

We used Mapchart software to analyze the location of *CsTPS* genes on cucumber chromosomes. The results showed that seven *CsTPS* genes were randomly located on six cucumber chromosomes (Fig. 4). Two genes (*CsTPS1* and *CsTPS2*) were located on

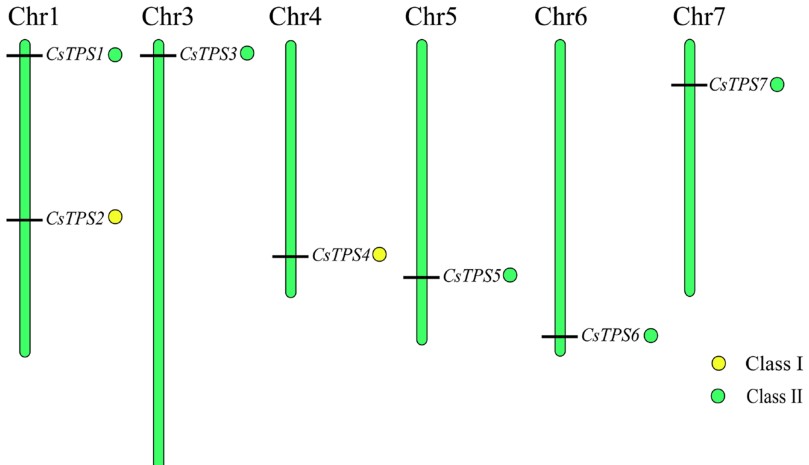

**Figure 4 TPS gene locations in cucumber chromosomes.** The chromosomes are represented by green bars.

chromosome 1, while only single genes existed on the other chromosomes. The majority of the *CsTPS* genes were located on the proximate or distal ends of the cucumber chromosomes (*Zhao et al., 2018*).

## *Cis*-acting element analysis of cucumber *TPS* genes

A total of 72 types of elements were found in the promoter regions of the *CsTPS* genes. All *CsTPS* genes contained CAAT and TATA boxes in the promoter regions, which were core and common promoter elements (*Zhou et al., 2017*) (Supplemental File 3). The *cis*-acting elements were identified and categorized as stress-related elements and plant hormone-responsive elements (*Zhu et al., 2019*). Stress-related elements, including MYB (stress response element), MYC (the recognition site of cold-resistant element), and ARE (anaerobic induction element) were mainly found in promoter regions of most *CsTPS* genes (*Zhou et al., 2017*). Some plant hormone-responsive elements, including ERE (ethylene-responsive element), ABRE (abscisic acid responsive element), TCA-element (salicylic acid-responsive element), and TGA-element (auxin-responsive element) (*Han et al., 2020*) were also found in most *CsTPS* genes (Figs. 5 and 6). Light responsive elements including Box4, AE-box, G-Box and GATA-motif were widely found in promoter regions of cucumber *TPS* genes (Fig. 5 and Supplemental File 3). Our results indicated that *TPS* genes played a significant role in stress, hormone and light response (*Zhao et al., 2018*).

## GO annotation of cucumber TPS proteins

We found that a majority of cucumber TPS proteins participated in UDP-glycosyltransferase activity, hydrolase activity, phosphatase activity, catalytic activity, and alpha-trehalose-phosphate synthase activity (Supplemental File 4). CsTPS2 and CsTPS4 were involved in transferase activity and phosphoric ester hydrolase activity. Cellular component analysis showed that most CsTPS proteins were located on the cytoplasm and cytosol. However, CsTPS2 and CsTPS4 were also located intracellularly. Biological

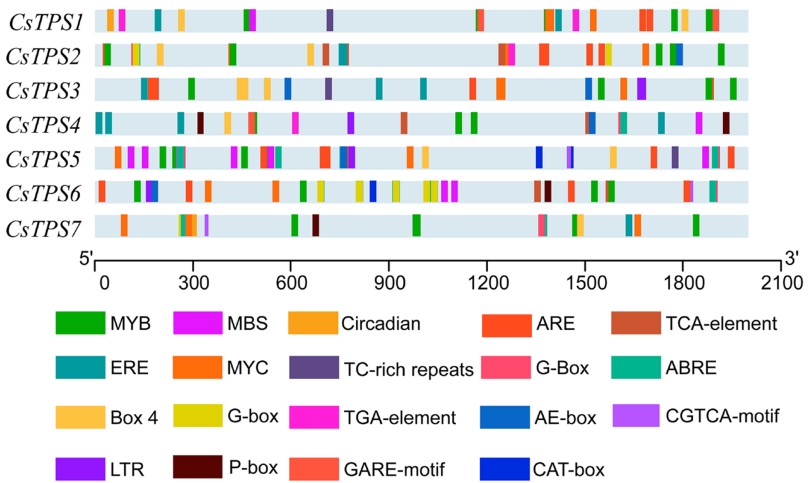

**Figure 5 The distribution of *cis*-elements in cucumber *TPS* genes.** Colored rectangles represent different *cis*-acting elements.

| | Stress-related elements | | | | | | Phytohormone-responsive elements | | | | | | |
|---|---|---|---|---|---|---|---|---|---|---|---|---|---|
| | MYB | MYC | MBS | ARE | LTR | TC-rich repeats | ABRE | CGTCA-motif | GARE-motif | P-box | TGA-element | ERE | TCA-element |
| *CsTPS1* | 6 | 3 | 1 | 2 | | 1 | | | 2 | | 2 | 2 | 1 |
| *CsTPS2* | 5 | 3 | | 5 | | | 1 | | | | 1 | 2 | 3 |
| *CsTPS3* | 4 | 4 | | 3 | 2 | 1 | | | 1 | | | 3 | |
| *CsTPS4* | 3 | | 1 | | 1 | | 1 | | 1 | 2 | 1 | 4 | 2 |
| *CsTPS5* | 4 | 2 | 5 | 7 | 1 | 1 | 3 | 2 | | | | 1 | |
| *CsTPS6* | 4 | 2 | 2 | 4 | 1 | | 11 | 1 | | 1 | | | 2 |
| *CsTPS7* | 5 | 3 | | | | | 2 | 1 | | 1 | | 1 | |

**Figure 6 The number of *Cis*-acting elements in cucumber *TPS* genes.**

process analysis indicated that most CsTPS proteins were involved in various biological processes, including metabolism, biosynthesis, cellular processes, and development. Some CsTPS proteins also participated in the stress response, signal transduction, post-embryonic development, and seed, fruit and reproductive system development.

### Expression analysis of cucumber *TPS* genes in different tissues

We measured the expression of *CsTPS* genes in root, stem and leaf using qRT-PCR to determine the expression specificity of cucumber *TPS* genes in different tissues. The expression of *CsTPS* genes was detected in the root, stem and leaf (Fig. 7) and their expressions were high, moderate, and low in the root, leaf and stem, respectively. Our results indicated that *CsTPS* genes may play a specific role in the growth of cucumber seedlings.

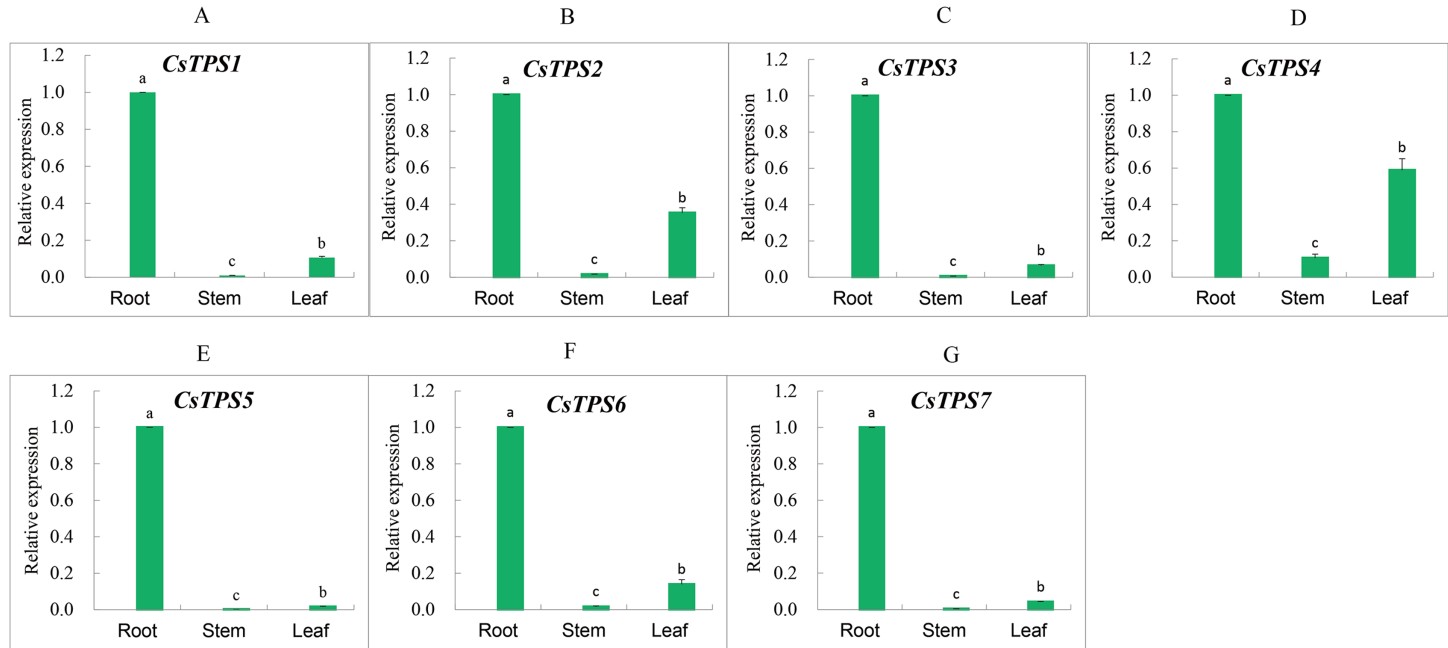

**Figure 7  Expression levels of *CsTPS* genes in root, stem and leaf.** The expression patterns of *CsTPS1–CsTPS7* in different tissues are shown in A–G, respectively. Error bars represent the standard error of three replicates. The relative expression of each gene in different tissues is expressed as mean ± SE (*n* = 3). Bars with different lowercase letters were significantly different by Duncan's multiple range tests (*p* < 0.05).

## Expression analysis of cucumber *TPS* genes under different treatments

We conducted qRT-PCR experiments and drew a cluster heatmap to determine the expression patterns of *CsTPS* genes under various treatments (Fig. 8). Our results indicated different patterns for the expression of *CsTPS* genes with various treatments. Under PEG, the expression of *CsTPS3* was up-regulated significantly and peaked at 24 h and *CsTPS2* and *CsTPS5* were up-regulated slightly. However, *CsTPS1*, *CsTPS4*, *CsTPS6* and *CsTPS7* were down-regulated and reached their lowest expression levels at 24 h. The expression of *CsTPS3* and *CsTPS4* was up-regulated significantly with NaCl treatment and reached the highest expression level at 24 h; the expression of *CsTPS2* and *CsTPS7* also increased. However, *CsTPS1* and *CsTPS6* were down-regulated with the NaCl treatment and reached the lowest expression level at 24 h. Under $H_2O_2$ treatment, *CsTPS1* and *CsTPS5* were activated while others genes were inhibited (*CsTPS2, CsTPS3, CsTPS4* and *CsTPS6*) or did not show obvious trends (*CsTPS7*). Under mannitol treatment, *CsTPS3* and *CsTPS7* were up-regulated with the highest expression level at 24 h and *CsTPS1* and *CsTPS4* were slightly induced. Conversely, the mannitol treatment caused a large decline in the expression of *CsTPS2*, *CsTPS5* and *CsTPS6*. Under IAA treatment, *CsTPS3* and *CsTPS4* were up-regulated significantly, whereas others genes showed no clear trends. *CsTPS3* and *CsTPS4* showed a strong expression under sucrose treatment, whereas *CsTPS1* and *CsTPS7* were induced slightly by sucrose. *CsTPS2*, *CsTPS5* and *CsTPS6* were inhibited by sucrose.
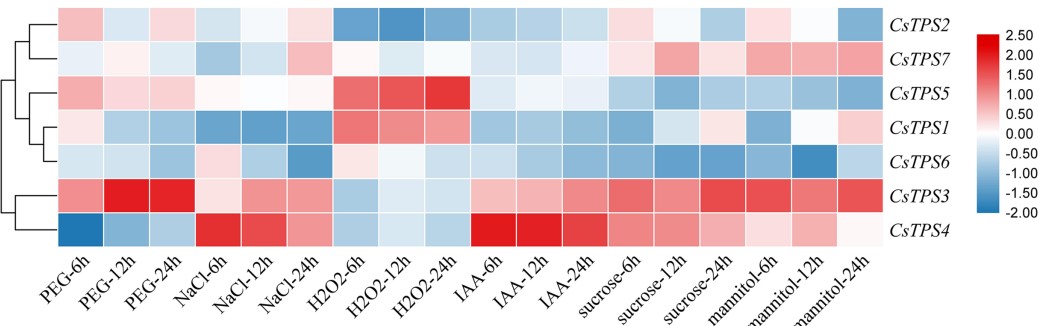

**Figure 8** Expression levels of *CsTPS* genes under PEG, NaCl, H₂O₂, IAA, mannitol and sucrose treatments. Seedlings were treated with 8% (w/v) PEG, 50 mM NaCl, 1 μM IAA, 8% (w/v) H₂O₂, 50 mM sucrose and 50 mM mannitol. The color scale represents fold changes normalized by log2 transformed data. Red represents up- regulated genes and blue represents down-regulated genes.

These results showed that *CsTPS* genes may be involved in stress, hormone, and sugar responses.

## DISCUSSION

Trehalose-6-phosphate synthase (TPS) is vital for the stress response and trehalose metabolism. Genes encoding TPS have been identified in many plants in the form of a gene family (*Avonce et al., 2004*). Previous studies have found 11, 11, 11, 8, 20, 13, 12, 10, 9 and 53 *TPS* genes in *Arabidopsis* (*Yang et al., 2012*), rice (*Zang et al., 2011*), pepper (*Wei et al., 2016*), potato (*Xu et al., 2017*), soybean (*Xie, Wang & Huang, 2014*), apple (*Du et al., 2017*), winter wheat (*Xie et al., 2015*), tomato (*Chen & Lou, 2017*), sugarcane (*Hu et al., 2020*), and cotton (*Mu et al., 2016*), respectively. However, the *TPS* gene family in cucumber has not been well-studied. We identified seven *TPS* genes in the cucumber genome, which were randomly located in six chromosomes of cucumber. Differences in the genome size of different species may cause variations in the number of members of the *TPS* family (*Xie et al., 2018*).

Cucumber *TPS* genes were divided into two subfamilies: Class I (*CsTPS2* and *CsTPS4*) and Class II, which was consistent with the classification in *Arabidopsis* (*Yang et al., 2012*), rice (*Zang et al., 2011*) and pepper (*Wei et al., 2016*). We found that four (*AtTPS1-AtTPS4*), one (*OsTPS1*) and three (*GaTPS1-GaTPS3*) *TPS* genes belonged to Class I in *Arabidopsis*, rice and pepper, respectively. Class I genes in our study had 16–17 introns, while Class II genes had two introns, with the exception of *CsTPS7*. Motif analysis showed that Class II genes possessed all 10 conserved motifs, whereas Class I genes lacked motif 8 and 9 in cucumber. These results were supported by results from earlier studies in *Arabidopsis* (*Yang et al., 2012*), rice (*Zang et al., 2011*) and cotton (*Mu et al., 2016*). Class I genes possessed 16 introns in *Arabidopsis* and rice, whereas Class II genes had two introns. Class I genes in cotton had more introns than Class II genes and lacked motif 8 (*Mu et al., 2016*). According to a previous study, three mechanisms (exon/intron gain/loss, exonization/pseudoexonization and insertion/deletion) may cause the difference in gene structure (*Xu et al., 2012*). These studies illustrated that the two subfamilies may

have experienced functional differentiation during evolution (*Xie et al., 2018*). Domain analysis indicated that all cucumber *TPS* genes possessed a TPS domain at the N-terminal and a TPP domain at the C-terminal, which was consistent with results from earlier studies in *Arabidopsis* (*Yang et al., 2012*), pepper (*Wei et al., 2016*) and apple (*Du et al., 2017*). CsTPS3, CsTPS5 and CsTPS6 proteins all contained a Hydrolase-3 (Pfam: PF08282) domain. However, some *TPS* genes lacked either a TPS domain or a TPP domain; *GrTPS6*, *GhTPS4* and *GhTPS9* genes in cotton lacked a TPP domain (*Mu et al., 2016*) and the loss of the domain may be the result of evolution. Evolutionary analysis showed that 49 TPS proteins from *Arabidopsis*, rice, cucumber and soybean were divided into two subfamilies and they were further classified into seven subgroups, similar to the classifications from a previous study (*Xie et al., 2015*). The *CsTPS* proteins were grouped together with at least one TPS protein from other species in the evolutionary tree, indicating that TPS proteins from different species had similar functions (*Zhou et al., 2017*).

Cis-acting elements were involved in the regulation of gene expression. Certain transcription factors have been shown to be activated and combined with *cis*-acting elements to activate the expression of stress-related genes when plants were exposed to adverse conditions (*Hadiarto & Tran, 2011*). We found that some elements related to stress (MBS, LTR, ARE), hormones (ABRE, ERE, TCA-element), and light response (AE-box, Box 4, TCT-motif) existed widely in promoter regions of most cucumber *TPS* genes. These elements consistently appeared in potato *TPS* genes (*Xu et al., 2017*), indicating that *TPS* genes may be involved in the stress, hormone, and light responses. The overexpression of *TPS* genes in *Arabidopsis*, rice, and potato improved their stress tolerance (*Avonce et al., 2004*; *Li et al., 2011*; *Kondrák et al., 2011*). Our study showed that *CsTPS3* was significantly induced by drought stress, which was consistent with a study in *Arabidopsis* (*Avonce et al., 2004*). *CsTPS3* and *CsTPS4* showed strong expression under salt stress, which was in agreement with the results of *OsTPS1* (*Li et al., 2011*). *CsTPS1* and *CsTPS5* showed strong expression under oxidative stress. *CsTPS3* and *CsTPS7* were clearly induced by osmotic stress by mannitol, which coincides with the results from cotton and watermelon (*Mu et al., 2016*). *CsTPS3* was induced significantly by drought, salt and osmotic stresses, indicating that *CsTPS3* may be more sensitive to various abiotic stress than other *CsTPS* genes. Plant hormones play an important role in signal transduction, plant growth, and development. We found that *CsTPS3* and *CsTPS4* were up-regulated by IAA, while other *CsTPS* genes were down-regulated by IAA, which were similar to the results for potato (*Xu et al., 2017*). GO analysis indicated that CsTPS proteins may participate in the response to stress, which further supported our results.

Studies have shown that *AtTPS1* participated in embryonic development and vegetative growth through the ABA mechanism and sugar metabolism (*Gómez et al., 2010*). *AtTPS1* played an essential role in regulating sugar signaling (*Avonce et al., 2004*) and *Arabidopsis TPSs* could be repressed or induced by sugar. We found that *CsTPS3* and *CsTPS4* were significantly induced by sucrose, while *CsTPS2*, *CsTPS5* and *CsTPS6* were repressed by sucrose. *CsTPS3* was highly sensitive to abiotic stress (except oxidative stress), hormone,

and sucrose treatments. In contrast, abiotic stress, hormone, and sucrose treatments caused a large decline in the expression of *CsTPS6*. The expression analysis of *CsTPS* genes in various tissues showed that *CsTPS* genes were expressed in the root, stem, and leaf and had the highest expression levels in the root. The expression of most *TPS* genes in cotton was induced by low temperature, salt and drought, while their expression patterns were different (*Mu et al., 2016*). In winter wheat, *TaTPS1* and *TaTPS3* expression was up-regulated under a freeze treatment (−20 °C) (*Xie et al., 2015*). In potato, the expression and patterns of *StTPS* genes were regulated by different stresses (salt, heat and osmotic) and hormones (IAA, ABA and GA$_3$) (*Xu et al., 2017*). In maize, the expression of *ZmTPS* genes was induced under salt and low temperature stresses (*Jiang et al., 2010*). In tomato, the expression of *SlTPS1* was inhibited by red and blue light (*Chen & Lou, 2017*). These results indicate that *TPS* genes may play critical roles in responding to stress, hormones, and light.

Trehalose may be involved in plant stress resistance including salt, cold, drought, and heavy metal. TPS is a key enzyme of trehalose metabolism with an essential role in plant stress resistance. Related studies have shown that the overexpression of *TPS* genes improved the tolerance of plants under unfavorable conditions (*Li et al., 2011*; *Kondrák et al., 2011*; *Liu et al., 2019b*). Only one or two genes classified as Class I encoded active trehalose-6-phosphate synthase (TPS) in most species. In rice, only proteins encoded by *OsTPS1* had TPS activity and all TPS proteins had no TPP activity (*Zang et al., 2011*). In *Arabidopsis*, only AtTPS1 (encoded by Class I genes) had TPS activity but no TPP activity, while TPS proteins encoded by Class II genes and remaining Class I genes had neither TPS nor TPP activity (*Yang et al., 2012*). In maize, ZmTPS1 possessed TPS activity (*Jiang et al., 2010*). We believe that CsTPS2 and CsTPS4 have TPS activity, but this requires further experimental support. Related studies have indicated that the accumulation of harmful mutations and changes in protein conformation may lead to the loss of TPS activity (*Yang et al., 2012*; *Vandesteene et al., 2010*). Previous studies have shown that the trehalose pathway was essential in regulating the use and distribution of sucrose, coordinating source-sink relation, the effective utilization of carbohydrates (*Schluepmann et al., 2003*), and improving crop yield (*Paul, Watson & Griffiths, 2020*). Studies in cucumber fruit showed that there was a strong correlation between T6P and sucrose (*Zhang et al., 2015*). The role of the trehalose pathway in reproductive growth and sugar metabolism should be the focus of future research.

## CONCLUSION

We identified seven *TPS* genes in the cucumber genome and analyzed their physicochemical properties, gene structures, domains, conserved motifs, evolutionary relationships, gene locations, *cis*-elements, GO analysis and expression patterns. Our results demonstrated that cucumber *TPS* genes play an important role in the response to stresses, sucrose, and phytohormones. Our study provides reference information for future studies of the mechanisms of TPS proteins on the growth, development, stress-resistance and trehalose pathway in cucumber.

### Funding

This work was supported by the National Key Research and Development Program (2018YFD1000800); the National Natural Science Foundation of China (Nos. 31860568, 31560563 and 31160398); the Research Fund of Higher Education of Gansu, China (No. 2018C-14); the Post-Doctoral Foundation of China (Nos. 20100470887 and 2012T50828) and the Natural Science Foundation of Gansu Province, China (Nos. 1606RJZA073 and 1606RJZA077). The funders had no role in study design, data collection and analysis, decision to publish, or preparation of the manuscript.

### Grant Disclosures

The following grant information was disclosed by the authors:
National Key Research and Development Program: 2018YFD1000800.
National Natural Science Foundation of China: 31860568, 31560563 and 31160398.
Research Fund of Higher Education of Gansu, China: 2018C-14.
Post-Doctoral Foundation of China: 20100470887 and 2012T50828.
Natural Science Foundation of Gansu Province, China: 1606RJZA073 and 1606RJZA077.

### Competing Interests

The authors declare that they have no competing interests.

### Author Contributions

- Yuanyuan Dan conceived and designed the experiments, performed the experiments, analyzed the data, prepared figures and/or tables, authored or reviewed drafts of the paper, and approved the final draft.
- Yuan Niu conceived and designed the experiments, authored or reviewed drafts of the paper, and approved the final draft.
- Chunlei Wang analyzed the data, authored or reviewed drafts of the paper, and approved the final draft.
- Mei Yan performed the experiments, authored or reviewed drafts of the paper, and approved the final draft.
- Weibiao Liao conceived and designed the experiments, analyzed the data, authored or reviewed drafts of the paper, and approved the final draft.

### Data Availability

The raw measurements are available in the Supplemental File.

### Supplemental Information

Supplemental information for this article can be found online at http://dx.doi.org/10.7717/peerj.11398#supplemental-information.

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
