# Peer review of "Genome-wide identification and expression analysis of the trehalose-6-phosphate synthase (TPS) gene family in cucumber (Cucumis sativus L.)"

_PeerJ, doi:10.7717/peerj.11398_

## Round 0.1 · original submission · Major Revisions

Please provide a comprehensively revised version addressing the editorial comments and a detailed rebuttal letter.

Reviewer 1 ·

Basic reporting

The manuscript shows the identification, characterization, and expression analysis of Trehalose-6-phosphate (TPS) genes of cucumber under different stress conditions. The manuscript structure is acceptable, however, there are points that should be check.

Experimental design

The authors mention that they used three technical replications (line 161 "Each reaction was repeated three times") for qRT-PCR-, but they do not comment how many biological replicates and how many plants they used for the analysis. This information is essential to support the statistical analysis and conclusions. The authors must add this information to the manuscript.

Line 87-95 Identification of TPS candidate sequences is not clear enough, it is suggested to define clearly the characteristics of TPS candidates, what were the characteristic domains or sequences that were searched in the HMMsearch analysis.
Line 91-92 How do you define an uncertain sequence? What was the goal of re-examining the sequences in Pfam and CDD?

NJ method for reconstruct the evolutionary relationship is weak method. ML o Bayesian methods would be preferred.

It is not clear why using root as calibration sample to calculate the relative gene expression (Fold change) in the different tissues. If the authors determine the relative gene expression only using the reference gene (CsActin) would give a better picture of how these genes are expressed in the tissues.

Validity of the findings

Line 209 It is not clear why the sequences were divided into six subgroups.

Lines 268-269 It is not clear why the authors indicate that CsTPS genes are involved in cucumber growth and development, they evaluated the expression at only a developmental stage.

The expression analysis of TPS genes in response to different types of stress shows interesting information, however, the discussion of these results is weak. It is suggested to perform a hierarchical clustering of the results of gene expression to identify common expression patterns, likewise, the authors must deepen the discussion of these results.

Lines 343-345 Is there previous evidence that plants have a low resistance to oxidative stress? If so, the authors should include references to complement this part.

Additional comments

For a better understanding of data presented, the authors should improve figure 3. It is suggested to add information that indicated which motifs represent each characteristic domains of the TPS proteins.

Line 279 This sentence is no clear.

·

Basic reporting

The sentence of the whole article is smooth, but there are still a few sentences that need to be further revised. It is suggested to ask foreign teachers to help modify it.

Experimental design

no comment

Validity of the findings

no comment

Additional comments

1. It is too simple to describe the expression patterns of genes under different stresses. The expression patterns of all genes under a single stress condition should be analyzed one by one. Then, in addition to the induction of genes which are more sensitive to stress. The author should express the content of the chart in the expression analysis more in words.
2. The discussion section needs to focus on one or two topics rather than on every study result. for example, focus on the expression pattern of one gene in different crops.
3. The study of transcription factors should be from the aspects of identification, chromosome location, phylogeny, structure, expression patterns, cis-elements in the promoter, and so on.
4. The full description of the cis-acting element can be placed in the test method and does not need to be described in the study result.

Reviewer 3 ·

Basic reporting

Improvements in English are required. Most significantly the paper does not consider the full literature regarding role of the trehalose pathway. See my fuller comments further down.

Experimental design

Seedlings only are used for expression analysis, yet the trehalose pathway may be particularly important in reproductive development, so the full picture does not emerge.

Validity of the findings

See above regarding comments on expression analysis. Critical function is overclaimed based on expression analysis in seedlings.

Additional comments

Genome-wide identification and expression analysis of the trehalose-6-phosphate synthase (TPS) gene family in cucumber (Cucumis sativus L.). Dan et al.
The paper documents information based on available bioinformatics data and expression analysis to assemble an analysis of the trehalose phosphate synthase (TPS) gene family in cucumber.
The writing is largely quite good and clear throughout with only a few language errors (see comments at end for examples of where improvements are necessary). However, consideration of the trehalose field in the paper is very limited and does not represent the current status of knowledge and sense of momentum that exists in the research area. In the Introduction and throughout the paper trehalose 6-phosphate (T6P) as critical component of the trehalose pathway is barely considered except for one reference to TPS1-T6P signaling pathway in tomato in the Introduction (Chen et al. 2017). There is a published example in cucumber that is not cited (Zhang et al. 2015 T6P and SNF-1 related protein kinase are involved in the first fruit inhibition of cucumber Journal of Plant Physiology 177, 110-120) which provides valuable context. The authors should refer to this and include it in interpreting their own results. T6P has been implicated in the regulation of sucrose homeostasis and in integrating sucrose supply with growth and development and coordination of source-sink relations in plants and crops. The trehalose pathway has been shown to be involved in improving stress resilience, but this may be through T6P rather than trehalose i.e. in adjusting sugar levels e.g. Nuccio et al. 2015 doi.org/10.1038/nbt.3277; Griffiths et al. 2016 doi.org/10.1038/nature20591. For an overview see T6P signalling and impact on crop yield doi.org.10.1042/BST20200286. As the trehalose pathway is so strongly linked to sugar metabolism, it is a pity that sucrose was not used in the studies that determined gene expression responses to different to different factors. TPSs of Arabidopsis can be sugar repressed or sugar induced. A major function of the trehalose pathway is in reproductive development and it is a shame again that expression levels of the TPSs were not determined during reproductive development and in fruit as this where the trehalose pathway may have a particularly strong role in yield formation in cucumber and in crops generally. However, 200 µmol m-2 s-1 irradiance may be too low for fruit formation. Cucumber requires sunny conditions, particularly for productive fruit formation. Do measurements of gene expression in cucumber seedlings grown at 200 µmol m-2 s-1 irradiance give a true picture for the crop?
The role of trehalose as stress protection compound is possible, but trehalose doesn’t accumulate to high enough levels to perform a major role of stress protectant and mode of action of trehalose is less clear than that of T6P including in cucumber (Zhang et al. 2015). The paper considers stress in terms of a more extremophile response whereas ongoing mild stress interspersed with moderate and severe stress plus periods of recovery is more of a reality for much of agriculture. Fruit production in cucumber will be the most stress-sensitive developmental phase from flowering to harvest, yet this aspect is not considered because seedlings only are used for gene expression analysis.
Line 354 and in the conclusion. The paper states that critical roles for TPSs have been established. Yet there is no direct evidence in this paper that TPSs perform critical roles in cucumber. This can only be assessed based on studies in other species. Expression studies are not enough to confirm critical function.
Line 368 speculation that TPS2 and TPS4 have catalytic activity. This is based I guess on the assumption that class I TPSs have catalytic activity? No direct evidence is supplied in the paper, though. How could this conclusion be better supported?
“Plant stress resist ability” line 360/ 361, replace with “plant stress resistance”.
In the Introduction are osmosis and superoxide adverse environments?

---

## Round 0.2 · Minor Revisions

Thanks for addressing the minor revisions requested. Now your manuscript is almost ready to be accepted in PeerJ.

A Section Editor has made the following comments. Please address the in a minor revision.

"In reading the first few paragraphs there were a number of grammar issues which needed clarity. PeerJ does not perform copyediting as part of the standard service, please ensure that the English language in this submission meets journal standards; this includes use of clear and unambiguous text which is grammatically correct, and conforms to professional standards of courtesy and expression.

Since the manuscript addressed the expression of different levels of development, tissue, and condition a need for the introduction of ontology annotations is needed. Journal manuscripts are often scanned by text-mining software that locates and extracts core data elements, like gene function. Adding standard ontology terms, such as the Gene Ontology (GO, geneontology.org) or others from the OBO foundry (obofoundry.org) can enhance the recognition of your contribution and description. This will also make human curation of literature easier and more accurate. None of this was visible.

Though the TPS genes were apparently characterized, there was no evidence of the actual sequences for the genes of study. Were these deposited to the NCBI resource, or are they extractable from a database version of the genome for cucumber?

The manuscript requires additional revisions."

Reviewer 1 ·

Basic reporting

I consider that the authors answered the observations and comments made. The information they added helped to complement the conclusions presented in this new version manuscript.

Experimental design

No comment.

Validity of the findings

No comment.

Additional comments

No comment.

·

Basic reporting

no comment

Experimental design

no comment

Validity of the findings

no comment

Additional comments

The author has revised the manuscript as required. it can be acceptable for publication

Reviewer 3 ·

Basic reporting

The paper is much improved in revision

Experimental design

Fine

Validity of the findings

Ok

Additional comments

The paper is much improved after revision

---

## Round 0.3 · accepted · Accept

Thanks for addressing all the revisions and corrections requested. Now your manuscript is accepted in PeerJ.